# Distributionally Robust Optimization and Generalization in Kernel Methods

**Matthew Staib**
MIT CSAIL
mstaib@mit.edu

**Stefanie Jegelka**
MIT CSAIL
stefje@csail.mit.edu

## Abstract

Distributionally robust optimization (DRO) has attracted attention in machine learning due to its connections to regularization, generalization, and robustness. Existing work has considered uncertainty sets based on $\phi$-divergences and Wasserstein distances, each of which have drawbacks. In this paper, we study DRO with uncertainty sets measured via maximum mean discrepancy (MMD). We show that MMD DRO is roughly equivalent to regularization by the Hilbert norm and, as a byproduct, reveal deep connections to classic results in statistical learning. In particular, we obtain an alternative proof of a generalization bound for Gaussian kernel ridge regression via a DRO lense. The proof also suggests a new regularizer. Our results apply beyond kernel methods: we derive a generically applicable approximation of MMD DRO, and show that it generalizes recent work on variance-based regularization.

## 1 Introduction

Distributionally robust optimization (DRO) is an attractive tool for improving machine learning models. Instead of choosing a model $f$ to minimize empirical risk $\mathbb{E}_{x \sim \hat{\mathbb{P}}_n}[\ell_f(x)] = \frac{1}{n} \sum_i \ell_f(x_i)$, an adversary is allowed to perturb the sample distribution within a set $\mathcal{U}$ centered around the empirical distribution $\hat{\mathbb{P}}_n$. DRO seeks a model that performs well regardless of the perturbation: $\inf_f \sup_{\mathbb{Q} \in \mathcal{U}} \mathbb{E}_{x \sim \mathbb{Q}}[\ell_f(x)]$. The induced robustness can directly imply generalization: if the data that forms $\hat{\mathbb{P}}_n$ is drawn from a population distribution $\mathbb{P}$, and $\mathcal{U}$ is large enough to contain $\mathbb{P}$, then we implicitly optimize for $\mathbb{P}$ too and the DRO objective value upper bounds out of sample performance. More broadly, robustness has gained attention due to adversarial examples [17, 39, 26]; indeed, DRO generalizes robustness to adversarial examples [35, 36].

In machine learning, the DRO uncertainty set $\mathcal{U}$ has so far always been defined as a $\phi$-divergence ball or Wasserstein ball around the empirical distribution $\hat{\mathbb{P}}_n$. These choices are convenient, due to a number of structural results. For example, DRO with $\chi^2$-divergence is roughly equivalent to regularizing by variance [18, 22, 31], and the worst case distribution $\mathbb{Q} \in \mathcal{U}$ can be computed exactly in $O(n \log n)$ [37]. Moreover, DRO with Wasserstein distance is asymptotically equivalent to certain common norm penalties [15], and the worst case $\mathbb{Q} \in \mathcal{U}$ can be computed approximately in several cases [28, 14]. These structural results are key, because the most challenging part of DRO is solving (or bounding) the DRO objective.

However, there are substantial drawbacks to these two types of uncertainty sets. Any $\phi$-divergence uncertainty set $\mathcal{U}$ around $\hat{\mathbb{P}}_n$ contains only distributions with the same (finite) support as $\hat{\mathbb{P}}_n$. Hence, the population $\mathbb{P}$ is typically *not* in $\mathcal{U}$, and so the DRO objective value cannot directly certify out of sample performance. Wasserstein uncertainty sets do not suffer from this problem. But, they are more computationally expensive, and the above key results on equivalences and computation need nontrivial assumptions on the loss function and the specific ground distance metric used.

In this paper, we introduce and develop a new class of DRO problems, where the uncertainty set $\mathcal{U}$ is defined with respect to the maximum mean discrepancy (MMD) [19], a kernel-based distance between distributions. MMD DRO complements existing approaches and avoids some of their drawbacks, e.g., unlike $\phi$-divergences, the uncertainty set $\mathcal{U}$ will contain $\mathbb{P}$ if the radius is large enough.

First, we show that MMD DRO is roughly equivalent to regularizing by the Hilbert norm $\|\ell_f\|_{\mathcal{H}}$ of the loss $\ell_f$ (not the model $f$). While, in general, $\|\ell_f\|_{\mathcal{H}}$ may be difficult to compute, we show settings in which it is tractable. Specifically, for kernel ridge regression with a Gaussian kernel, we prove a bound on $\|\ell_f\|_{\mathcal{H}}$ that, as a byproduct, yields generalization bounds that match (up to a small constant) the standard ones. Second, beyond kernel methods, we show how MMD DRO generalizes variance-based regularization. Finally, we show how MMD DRO can be efficiently approximated empirically, and in fact generalizes variance-based regularization.

Overall, our results offer deeper insights into the landscape of regularization and robustness approaches, and a more complete picture of the effects of different divergences for defining robustness. In short, our contributions are:

1. We prove fundamental structural results for MMD DRO, and its rough equivalence to penalizing by the Hilbert norm of the loss.
2. We give a new generalization proof for Gaussian kernel ridge regression by way of DRO. Along the way, we prove bounds on the Hilbert norm of products of functions that may be of independent interest.
3. Our generalization proof suggests a new regularizer for Gaussian kernel ridge regression.
4. We derive a computationally tractable approximation of MMD DRO, with application to general learning problems, and we show how the aforementioned approximation generalizes variance regularization.

## 2 Background and Related Work

Distributionally robust optimization (DRO) [16, 2], introduced by Scarf [32], asks to not only perform well on a fixed problem instance (parameterized by a distribution), but simultaneously for a range of problems, each determined by a distribution in an *uncertainty set* $\mathcal{U}$. This results in more robust solutions. The uncertainty set plays a key role: it implicitly defines the induced notion of robustness. The DRO problem we address asks to learn a model $f$ that solves

$$\text{(DRO)} \qquad \inf_f \sup_{\mathbb{Q} \in \mathcal{U}} \mathbb{E}_{x \sim \mathbb{Q}}[\ell_f(x)], \qquad (1)$$

where $\ell_f(x)$ is the loss incurred under prediction $f(x)$.

In this work, we focus on *data-driven DRO*, where $\mathcal{U}$ is centered around an empirical sample $\hat{\mathbb{P}}_n = \frac{1}{n}\sum_{i=1}^n \delta_{x_i}$, and its size is determined in a data-dependent way. Data-driven DRO yields a natural approach for certifying out-of-sample performance.

**Principle 2.1** (DRO Generalization Principle)**.** *Fix any model $f$. Let $\mathcal{U}$ be a set of distributions containing $\hat{\mathbb{P}}_n$. Suppose $\mathcal{U}$ is large enough so that, with probability $1 - \delta$, $\mathcal{U}$ contains the population $\mathbb{P}$. Then with probability $1 - \delta$, the population loss $\mathbb{E}_{x \sim \mathbb{P}}[\ell_f(x)]$ is bounded by*

$$\mathbb{E}_{x \sim \mathbb{P}}[\ell_f(x)] \ \leq \ \sup_{\mathbb{Q} \in \mathcal{U}} \mathbb{E}_{x \sim \mathbb{Q}}[\ell_f(x)]. \qquad (2)$$

Essentially, if the uncertainty set $\mathcal{U}$ is chosen appropriately, the corresponding DRO problem gives a high probability bound on population performance. The two key steps in using Principle 2.1 are **1.** arguing that $\mathcal{U}$ actually contains $\mathbb{P}$ with high probability (e.g. via concentration); **2.** solving the DRO problem on the right hand side, or an upper bound thereof.

In practice, $\mathcal{U}$ is typically chosen as a ball of radius $\epsilon$ around the empirical sample $\hat{\mathbb{P}}_n$: $\mathcal{U} = \{\mathbb{Q} : d(\mathbb{Q}, \hat{\mathbb{P}}_n) \leq \epsilon\}$. Here, $d$ is a discrepancy between distributions, and is of utmost significance: the choice of $d$ determines how large $\epsilon$ must be, and how tractable the DRO problem is.

In machine learning, two choices of the divergence $d$ are prevalent, $\phi$-divergences [1, 11, 22], and Wasserstein distance [28, 33, 6]. The first option, $\phi$-divergences, have the form $d_\phi(\mathbb{P}, \mathbb{Q}) =$

$\int \phi(d\mathbb{P}/d\mathbb{Q})\,d\mathbb{Q}$. In particular, they include the $\chi^2$-divergence, which makes the DRO problem equivalent to regularizing by variance [18, 22, 31]. Beyond better generalization, variance regularization has applications in fairness [20]. However, a major shortcoming of DRO with $\phi$-divergences is that the ball $\mathcal{U} = \{\mathbb{Q} : d_\phi(\mathbb{Q}, \mathbb{P}_0) \leq \epsilon\}$ only contains distributions $\mathbb{Q}$ whose support is contained in the support of $\mathbb{P}_0$. If $\mathbb{P}_0 = \hat{\mathbb{P}}_n$ is an empirical distribution on $n$ points, the ball $\mathcal{U}$ only contains distributions with the same finite support. Hence, the population distribution $\mathbb{P}$ typically cannot belong to $\mathcal{U}$, and it is not possible to certify out-of-sample perfomance by Principle 2.1. Though Principle 2.1 does not apply here, generalization bounds are still possible via other means [31].

The second option, Wasserstein distance, is based on a distance metric $g$ on the data space. The $p$-Wasserstein distance $W_p$ between measure $\mu, \nu$ is given by $W_p(\mu, \nu) = \inf\{\int g(x, y)^p \, d\gamma(x, y) : \gamma \in \Pi(\mu, \nu)\}^{1/p}$, where $\Pi(\mu, \nu)$ is the set of couplings of $\mu$ and $\nu$ [40]. Wasserstein DRO has a key benefit over $\phi$-divergences: the set $\mathcal{U} = \{\mathbb{Q} : W_p(\mathbb{Q}, \mathbb{P}_0) \leq \epsilon\}$ contains continuous distributions. Moreover, concentration results bounding $W_p(\mathbb{P}, \hat{\mathbb{P}}_n)$ with high probability are available for many settings, e.g. [13, 23, 34, 41]. However, Wasserstein distance is much harder to work with, and nontrivial assumptions are needed to derive the necessary structural and algorithmic results for solving the associated DRO problem. To our knowledge, in all Wasserstein DRO work so far, the ground metric $g$ is limited to slight variations of either a Euclidean or Mahalanobis metric [7, 8]. Such metrics may be a poor fit for complex data such as images or distributions. These assumptions restrict the extent to which Wasserstein DRO can utilize complex, nonlinear structure in the data.

**Maximum Mean Discrepancy (MMD).** MMD is a distance metric between distributions that leverages kernel embeddings. Let $\mathcal{H}$ be a reproducing kernel Hilbert space (RKHS) with kernel $k$ and norm $\|\cdot\|_{\mathcal{H}}$. MMD is defined as follows:

**Definition 2.1.** The *maximum mean discrepancy (MMD)* between distributions $\mathbb{P}$ and $\mathbb{Q}$ is

$$d_{\mathrm{MMD}}(\mathbb{P}, \mathbb{Q}) := \sup_{f \in \mathcal{H}: \|f\|_{\mathcal{H}} \leq 1} \mathbb{E}_{x \sim \mathbb{P}}[f(x)] - \mathbb{E}_{x \sim \mathbb{Q}}[f(x)]. \tag{3}$$

**Fact 2.1.** Define the mean embedding $\mu_{\mathbb{P}}$ of the distribution $\mathbb{P}$ by $\mu_{\mathbb{P}} = \mathbb{E}_{x \sim \mathbb{P}}[k(x, \cdot)]$. Then the MMD between distributions $\mathbb{P}$ and $\mathbb{Q}$ can be equivalently written

$$d_{\mathrm{MMD}}(\mathbb{P}, \mathbb{Q}) = \|\mu_{\mathbb{P}} - \mu_{\mathbb{Q}}\|_{\mathcal{H}}. \tag{4}$$

MMD and (more generally) kernel mean embeddings have been used in many applications, particularly in two- and one-sample tests [19, 21, 25, 9] and in generative modeling [12, 24, 38, 5]. We refer the interested reader to the monograph by Muandet et al. [30]. MMD admits efficient estimation, as well as fast convergence properties, which are of chief importance in our work.

**Further related work.** Beyond $\phi$-divergences and Wasserstein distances, work in operations research has considered DRO problems that capture uncertainty in moments of the distribution, e.g. [10]. These approaches typically focus on first- and second-order moments; in contrast, an MMD uncertainty set allows high order moments to vary, depending on the choice of kernel.

Robust and adversarial machine learning have strong connections to our work and DRO more generally. Robustness to adversarial examples [39, 17], where individual inputs to the model are perturbed in a small ball, can be cast as a robust optimization problem [26]. When the ball is a norm ball, this robust formulation is a special case of Wasserstein DRO [35, 36]. Xu et al. [42] study the connection between robustness and regularization in SVMs, and perturbations within a (possibly Hilbert) norm ball. Unlike our work, their results are limited to SVMs instead of general loss minimization. Moreover, they consider only perturbation of individual data points instead of shifts in the entire *distribution*. Bietti et al. [4] show that many regularizers used for neural networks can also be interpreted in light of an appropriately chosen Hilbert norm [3].

## 3 Generalization bounds via MMD DRO

The main focus of this paper is Distributionally Robust Optimization where the uncertainty set is defined via the MMD distance $d_{\mathrm{MMD}}$:

$$\inf_f \sup_{\mathbb{Q}:d_{\mathrm{MMD}}(\mathbb{Q}, \hat{\mathbb{P}}_n) \leq \epsilon} \mathbb{E}_{x \sim \mathbb{Q}}[\ell_f(x)]. \tag{5}$$

One motivation for considering MMD in this setting are its possible implications for Generalization. Recall that for the DRO Generalization Principle 2.1 to apply, the uncertainty set $\mathcal{U}$ must contain the population distribution with high probability. To ensure this, the radius of $\mathcal{U}$ must be large enough. But, the larger the radius, the more pessimistic is the DRO minimax problem, which may lead to over-regularization. This radius depends on how quickly $d_{\mathrm{MMD}}(\mathbb{P}, \hat{\mathbb{P}}_n)$ shrinks to zero, i.e., on the empirical accuracy of the divergence.

In contrast to Wasserstein distance, which converges at a rate of $O(n^{-1/d})$ [13], MMD between the empirical sample $\hat{\mathbb{P}}_n$ and population $\mathbb{P}$ shrinks as $O(n^{-1/2})$:

**Lemma 3.1** (Modified from [30], Theorem 3.4). *Suppose that $k(x,x) \leq M$ for all $x$. Let $\hat{\mathbb{P}}_n$ be an $n$ sample empirical approximation to $\mathbb{P}$. Then with probability $1 - \delta$,*

$$d_{\mathrm{MMD}}(\mathbb{P}, \hat{\mathbb{P}}_n) \leq 2\sqrt{\frac{M}{n}} + \sqrt{\frac{2\log(1/\delta)}{n}}. \tag{6}$$

The constant $M$ is dimension-independent for many common universal kernels, e.g. Gaussian, Laplace, and Matern kernels. With Lemma 3.1 in hand, we conclude a simple high probability bound on out-of-sample performance:

**Corollary 3.1.** *Suppose that $k(x,x) \leq M$ for all $x$. Set $\epsilon = 2\sqrt{M/n} + \sqrt{2\log(1/\delta)/n}$. Then with probability $1 - \delta$, we have the following bound on population risk:*

$$\mathbb{E}_{x \sim \mathbb{P}}[\ell_f(x)] \leq \sup_{\mathbb{Q}: d_{\mathrm{MMD}}(\mathbb{Q}, \hat{\mathbb{P}}_n) \leq \epsilon} \mathbb{E}_{x \sim \mathbb{Q}}[\ell_f(x)]. \tag{7}$$

We refer to the right hand side as the DRO adversary's problem. In the next section we develop results that enable us to bound its value, and consequently bound the DRO problem (5).

## 3.1 Bounding the DRO adversary's problem

The DRO adversary's problem seeks the distribution $\mathbb{Q}$ in the MMD ball so that $\mathbb{E}_{x \sim \mathbb{Q}}[\ell_f(x)]$ is as high as possible. Reasoning about the optimal worst-case $\mathbb{Q}$ is the main difficulty in DRO. With MMD, we take two steps for simplification. First, instead of directly optimizing over distributions, we optimize over their mean embeddings in the Hilbert space (described in Fact 2.1). Second, while the adversary's problem (7) makes sense for general $\ell_f$, we assume that the loss $\ell_f$ is in $\mathcal{H}$. In case $\ell_f \notin \mathcal{H}$, often $k$ is a universal kernel, meaning under mild conditions $\ell_f$ can be approximated arbitrarily well by a member of $\mathcal{H}$ [30, Definition 3.3].

With the additional assumption that $\ell_f \in \mathcal{H}$, the risk $\mathbb{E}_{x \sim \mathbb{P}}[\ell_f(x)]$ can also be written as $\langle \ell_f, \mu_{\mathbb{P}} \rangle_{\mathcal{H}}$. Then we obtain

$$\sup_{\mathbb{Q}: d_{\mathrm{MMD}}(\mathbb{Q}, \mathbb{P}) \leq \epsilon} \mathbb{E}_{x \sim \mathbb{Q}}[\ell_f(x)] \leq \sup_{\mu_{\mathbb{Q}} \in \mathcal{H}: \|\mu_{\mathbb{Q}} - \mu_{\mathbb{P}}\|_{\mathcal{H}} \leq \epsilon} \langle \ell_f, \mu_{\mathbb{Q}} \rangle_{\mathcal{H}}, \tag{8}$$

where we have an inequality because not every function in $\mathcal{H}$ is the mean embedding of some probability distribution. If $k$ is a characteristic kernel [30, Definition 3.2], the mapping $\mathbb{P} \mapsto \mu_{\mathbb{P}}$ is injective. In this case, the only looseness in the bound is due to discarding the constraints that $\mathbb{Q}$ integrates to one and is nonnegative. However it is difficult to constrain the mean embedding $\mu_{\mathbb{Q}}$ in this way as it is a function.

The mean embedding form of the problem is simpler to work with, and leads to further interpretations.

**Theorem 3.1.** *Let $\ell_f, \mu_{\mathbb{P}} \in \mathcal{H}$. We have the following equality:*

$$\sup_{\mu_{\mathbb{Q}} \in \mathcal{H}: \|\mu_{\mathbb{Q}} - \mu_{\mathbb{P}}\|_{\mathcal{H}} \leq \epsilon} \langle \ell_f, \mu_{\mathbb{Q}} \rangle_{\mathcal{H}} = \langle \ell_f, \mu_{\mathbb{P}} \rangle_{\mathcal{H}} + \epsilon \|\ell_f\|_{\mathcal{H}} = \mathbb{E}_{x \sim \mathbb{P}}[\ell_f(x)] + \epsilon \|\ell_f\|_{\mathcal{H}}. \tag{9}$$

*In particular, the adversary's optimal solution is $\mu_{\mathbb{Q}}^* = \mu_{\mathbb{P}} + \frac{\epsilon}{\|\ell\|_{\mathcal{H}}} \ell_f$.*

Combining Theorem 3.1 with equation (8) yields our main result for this section:

**Corollary 3.2.** *Let $\ell_f \in \mathcal{H}$, let $\mathbb{P}$ be a probability distribution, and fix $\epsilon > 0$. Then,*

$$\sup_{\mathbb{Q}: d_{\mathrm{MMD}}(\mathbb{P}, \mathbb{Q}) \leq \epsilon} \mathbb{E}_{x \sim \mathbb{Q}}[\ell_f(x)] \ \leq \ \mathbb{E}_{x \sim \mathbb{P}}[\ell_f(x)] + \epsilon \|\ell_f\|_{\mathcal{H}} \qquad \textit{and therefore} \tag{10}$$

$$\inf_f \sup_{\mathbb{Q}: d_{\mathrm{MMD}}(\mathbb{P}, \mathbb{Q}) \leq \epsilon} \mathbb{E}_{x \sim \mathbb{Q}}[\ell_f(x)] \ \leq \ \inf_f \mathbb{E}_{x \sim \mathbb{P}}[\ell_f(x)] + \epsilon \|\ell_f\|_{\mathcal{H}}. \tag{11}$$

Combining Corollary 3.2 with Corollary 3.1 shows that minimizing the empirical risk plus a norm on $\ell_f$ leads to a high probability bound on out-of-sample performance. This result is similar to results that equate Wasserstein DRO to norm regularization. For example, Gao et al. [15] show that under appropriate assumptions on $\ell_f$, DRO with a $p$-Wasserstein ball is asymptotically equivalent to $\mathbb{E}_{x \sim \hat{\mathbb{P}}_n}[\ell_f(x)] + \epsilon \|\nabla_x \ell_f\|_{\hat{\mathbb{P}}_n, q}$, where $\|\nabla_x \ell_f\|_{\hat{\mathbb{P}}_n, q} = \left(\frac{1}{n} \sum_{i=1}^n \|\nabla_x \ell_f(x_i)\|_*^q\right)^{1/q}$ measures a kind of $q$-norm average of $\|\nabla_x \ell_f(x_i)\|_*$ at each data point $x_i$ (here $q$ is such that $1/p + 1/q = 1$, and $\|\cdot\|_*$ is the dual norm of the metric defining the Wasserstein distance).

There are a few key differences between our result and that of Gao et al. [15]. First, the norms are different. Second, their result penalizes only the gradient of $\ell_f$, while ours penalizes $\ell_f$ directly. Third, except for certain special cases, the Wasserstein results cannot serve as a true upper bound; there are higher order terms that only shrink to zero as $\epsilon \to 0$. These higher order terms may not be so small: in high dimension $d$, the radius $\epsilon$ of the uncertainty set needed so that $\mathbb{P} \in \mathcal{U}$ shrinks very slowly, as $O(n^{-1/d})$ [13].

**Remark 3.1.** Theorem 3.1 and Corollary 3.2 require that $\ell_f$ is in the RKHS $\mathcal{H}$. Though this may seem restrictive, if the kernel $k$ is universal, as is the case for many kernels used in practice such as Gaussian and Laplace kernels, we can readily extend our results to all bounded continuous functions. Suppose $\ell_f$ is a bounded continuous function on a compact metric space $\mathcal{X}$. By definition (e.g. [30], Definition 3.3), if $k$ is a universal kernel on $\mathcal{X}$, then for any $\epsilon > 0$, there is some $\ell' \in \mathcal{H}$ with $\sup_{x \in \mathcal{X}} |\ell_f(x) - \ell'(x)| < \epsilon$. It follows that for any measure $\mathbb{P}$, we can bound the expectation of $\ell_f(x)$ by that of $\ell'$: $\mathbb{E}_{x \sim \mathbb{P}}[\ell_f(x)] < \mathbb{E}_{x \sim \mathbb{P}}[\ell'(x)] + \epsilon$. Then, we can apply our results to $\ell' \in \mathcal{H}$.

# 4  Connections to kernel ridge regression

After applying Corollary 3.2, we are interested in solving:

$$\inf_f \mathbb{E}_{x \sim \hat{\mathbb{P}}_n}[\ell_f(x)] + \epsilon \|\ell_f\|_{\mathcal{H}}. \tag{12}$$

Here, we penalize our model $f$ by $\|\ell_f\|_{\mathcal{H}}$. This looks similar to but is very different from the usual penalty $\|f\|_{\mathcal{H}}$ in kernel methods. In fact, Hilbert norms of function compositions such as $\ell_f$ pose several challenges. For example, $f$ and $\ell_f$ may not belong to the same RKHS – it is not hard to construct counterexamples, even when $\ell$ is merely quadratic. So, the objective (12) is not yet computational.

Despite these challenges, we next develop tools that will allow us to bound $\|\ell_f\|_{\mathcal{H}}$ and use it as a regularizer. These tools may be of independent interest to bound RKHS norms of composite functions (e.g., for settings as in [4]). Due to the difficulty of this task, we specialize to Gaussian kernels $k_\sigma(x, y) = \exp(-\|x - y\|^2/(2\sigma^2))$. Since we will need to take care regarding the bandwidth $\sigma$, we explicitly write it out for the inner product $\langle \cdot, \cdot \rangle_\sigma$ and norm $\|\cdot\|_\sigma$, of the corresponding RKHS $H_\sigma$.

To make the setting concrete, consider kernel ridge regression, with Gaussian kernel $k_\sigma$. As usual, we assume there is a simple target function $h$ that fits our data: $h(x_i) = y_i$. Then the loss $\ell_f$ of $f$ is $\ell_f(x) = (f(x) - h(x))^2$, so we wish to solve

$$\inf_f \mathbb{E}_{x \sim \hat{\mathbb{P}}_n}[(f(x) - h(x))^2] + \epsilon \|(f - h)^2\|_\sigma. \tag{13}$$

## 4.1  Bounding norms of products

To bound $\|(f - h)^2\|_\sigma$, it will suffice to bound RKHS norms of products. The key result for this subsection is the following deceptively simple-looking bound:

**Theorem 4.1.** *Let $f, g \in \mathcal{H}_\sigma$, that is, the RKHS corresponding to the Gaussian kernel $k_\sigma$ of bandwidth $\sigma$. Then, $\|fg\|_{\sigma/\sqrt{2}} \le \|f\|_\sigma \|g\|_\sigma$.*

Indeed, there are already subtleties: if $f, g \in \mathcal{H}_\sigma$, then, to discuss the norm of the product $fg$, we need to decrease the bandwidth from $\sigma$ to $\sigma/\sqrt{2}$.

We prove Theorem 4.1 via two steps. First, we represent the functions $f, g$, and $fg$ *exactly* in terms of traces of certain matrices. This step is highly dependent on the specific structure of the Gaussian kernel. Then, we can apply standard trace inequalities. Proofs of both results are given in Appendix B.

**Proposition 4.1.** *Let $f, g \in \mathcal{H}_\sigma$ have expansions $f = \sum_i a_i k_\sigma(x_i, \cdot)$ and $g = \sum_j b_j k_\sigma(x_j, \cdot)$. For shorthand denote by $z_i = \phi_{\sqrt{2}\sigma}(x_i)$ the (possibly infinite) feature expansion of $x_i$ in $\mathcal{H}_{\sqrt{2}\sigma}$. Then,*

$$\|fg\|^2_{\sigma/\sqrt{2}} = \mathrm{tr}(A^2 B^2), \quad \|f\|^2_\sigma = \mathrm{tr}(A^2), \quad \text{and} \quad \|g\|^2_\sigma = \mathrm{tr}(B^2),$$

*where $A = \sum_i a_i z_i z_i^T$ and $B = \sum_j a_j z_j z_j^T$.*

**Lemma 4.1.** *Let $X, Y$ be symmetric and positive semidefinite. Then $\mathrm{tr}(XY) \leq \mathrm{tr}(X)\,\mathrm{tr}(Y)$.*

With these intermediate results in hand, we can prove the main bound of interest:

*Proof of Theorem 4.1.* By Proposition 4.1, we may write

$$\|fg\|^2_{\sigma/\sqrt{2}} = \mathrm{tr}(A^2 B^2), \quad \|f\|^2_\sigma = \mathrm{tr}(A^2), \quad \text{and} \quad \|g\|^2_\sigma = \mathrm{tr}(B^2),$$

where $A = \sum_i a_i z_i z_i^T$ and $B = \sum_j b_j z_j z_j^T$ are chosen as described in Proposition 4.1. Since $A$ and $B$ are each symmetric, it follows that $A^2$ and $B^2$ are each symmetric and positive semidefinite. Then we can apply Lemma 4.1 to conclude that

$$\|fg\|^2_{\sigma/\sqrt{2}} = \mathrm{tr}(A^2 B^2) \leq \mathrm{tr}(A^2)\,\mathrm{tr}(B^2) = \|f\|^2_\sigma \|g\|^2_\sigma. \qquad \square$$

### 4.2 Implications: kernel ridge regression

With the help of Theorem 4.1, we can develop DRO-based bounds for actual learning problems. In this section we develop such bounds for Gaussian kernel ridge regression, i.e. problem (13).

For shorthand, we write $R_\mathbb{Q}(f) = \mathbb{E}_{x \sim \mathbb{Q}}[\ell_f(x)] = \mathbb{E}_{x \sim \mathbb{Q}}[(f(x) - h(x))^2]$ for the risk of $f$ on a distribution $\mathbb{Q}$. Generalization amounts to proving that the population risk $R_\mathbb{P}(f)$ is not too different than the empirical risk $R_{\hat{\mathbb{P}}_n}(f)$.

**Theorem 4.2.** *Assume the target function $h$ satisfies $\|h^2\|_{\sigma/\sqrt{2}} \leq \Lambda_{h^2}$ and $\|h\|_\sigma \leq \Lambda_h$. Then, for any $\delta > 0$, with probability $1 - \delta$, the following holds for all functions $f$ satisfying $\|f^2\|_{\sigma/\sqrt{2}} \leq \Lambda_{f^2}$ and $\|f\|_\sigma \leq \Lambda_f$:*

$$R_\mathbb{P}(f) \leq R_{\hat{\mathbb{P}}_n}(f) + \frac{2}{\sqrt{n}} \left(1 + \sqrt{\frac{\log(1/\delta)}{2}}\right) \left(\Lambda_{f^2} + \Lambda_{h^2} + 2\Lambda_f \Lambda_h\right). \tag{14}$$

*Proof.* We utilize the DRO Generalization Principle 2.1, By Lemma 3.1 we know that with probability $1 - \delta$, $d_{\mathrm{MMD}}(\hat{\mathbb{P}}_n, \mathbb{P}) \leq \epsilon$ for $\epsilon = (2 + \sqrt{2\log(1/\delta)})/\sqrt{n}$, since $k_\sigma(x, x) \leq M = 1$. Note the bandwidth $\sigma$ does not affect the convergence result. As a result of Lemma 3.1, with probability $1 - \delta$:

$$R_\mathbb{P}(f) = \mathbb{E}_{x \sim \mathbb{P}}[(f(x) - h(x))^2] \tag{15}$$

$$\overset{(a)}{\leq} \mathbb{E}_{x \sim \hat{\mathbb{P}}_n}[(f(x) - h(x))^2] + \epsilon \|(f - h)^2\|_{\sigma/\sqrt{2}} \tag{16}$$

$$\overset{(b)}{\leq} R_{\hat{\mathbb{P}}_n}(f) + \epsilon \left(\|f^2\|_{\sigma/\sqrt{2}} + \|h^2\|_{\sigma/\sqrt{2}} + 2\|fh\|_{\sigma/\sqrt{2}}\right) \tag{17}$$

$$\overset{(c)}{\leq} R_{\hat{\mathbb{P}}_n}(f) + \epsilon \left(\Lambda_{f^2} + \Lambda_{h^2} + 2\Lambda_f \Lambda_h\right), \tag{18}$$

where (a) is by Corollary 3.2, (b) is by the triangle inequality, and (c) follows from Theorem 4.1 and our assumptions on $f$ and $h$. Plugging in the bound on $\epsilon$ yields the result. $\qquad \square$

We placed different bounds on each of $f, h, f^2, h^2$ to emphasize the dependence on each. Since each is bounded separately, the DRO based bound in Theorem 4.2 allows finer control of the complexity of the function class than is typical. Since, by Theorem 4.1, the norms of $f^2$, $h^2$ and $fh$ are bounded by those of $f$ and $h$, we may also state Theorem 4.2 just with $\|f\|_\sigma$ and $\|h\|_\sigma$.

**Corollary 4.1.** *Assume the target function $h$ satisfies $\|h\|_\sigma \leq \Lambda$. Then, for any $\delta > 0$, with probability $1 - \delta$, the following holds for all functions $f$ satisfying $\|f\|_\sigma \leq \Lambda$:*

$$R_\mathbb{P}(f) \leq R_{\hat{\mathbb{P}}_n}(f) + \frac{8\Lambda^2}{\sqrt{n}} \left(1 + \sqrt{\frac{\log(1/\delta)}{2}}\right). \tag{19}$$

*Proof.* We reduce to Theorem 4.2. By Theorem 4.1, we know that $\|f^2\|_{\sigma/\sqrt{2}} \leq \|f\|_\sigma^2$, which may be bounded above by $\Lambda^2$ (and similarly for $h$). Therefore we can take $\Lambda_{f^2} = \Lambda_f^2 = \Lambda$ and $\Lambda_{h^2} = \Lambda_h^2 = \Lambda$ in Theorem 4.2. The result follows by bounding

$$\Lambda_{f^2} + \Lambda_{h^2} + 2\Lambda_f\Lambda_h \leq \Lambda^2 + \Lambda^2 + 2\Lambda \cdot \Lambda = 4\Lambda^2. \qquad \square$$

Generalization bounds for kernel ridge regression are of course not new; we emphasize that the DRO viewpoint provides an intuitive approach that also grants finer control over the function complexity. Moreover, our results take essentially the same form as the typical generalization bounds for kernel ridge regression, reproduced below:

**Theorem 4.3** (Specialized from [29], Theorem 10.7)**.** *Assume the target function $h$ satisfies $\|h\|_\sigma \leq \Lambda$. Then, for any $\delta > 0$, with probability $1 - \delta$, it holds for all functions $f$ satisfying $\|f\|_\sigma \leq \Lambda$ that*

$$R_\mathbb{P}(f) \leq R_{\hat{\mathbb{P}}_n}(f) + \tfrac{8\Lambda^2}{\sqrt{n}}\left(1 + \tfrac{1}{2}\sqrt{\tfrac{\log(1/\delta)}{2}}\right). \tag{20}$$

Hence, our DRO-based Theorem 4.2 evidently recovers standard results up to a universal constant.

## 4.3 Algorithmic implications

The generalization result in Theorem 4.3 is often used to justify penalizing by the norm $\|f\|_\sigma$, since it is the only part of the bound (other than the risk $R_{\hat{\mathbb{P}}_n}(f)$) that depends on $f$. In contrast, our DRO-based generalization bound in Theorem 4.2 is of the form

$$R_\mathbb{P}(f) - R_{\hat{\mathbb{P}}_n}(f) \leq \epsilon\left(\|f^2\|_{\sigma/\sqrt{2}} + \|h^2\|_{\sigma/\sqrt{2}} + 2\|f\|_\sigma\|h\|_\sigma\right), \tag{21}$$

which depends on $f$ through both norms $\|f\|_\sigma$ and $\|f^2\|_{\sigma/\sqrt{2}}$. This bound motivates the use of both norms as regularizers in kernel regression, i.e. we would instead solve

$$\inf_{f \in \mathcal{H}_\sigma} \mathbb{E}_{(x,y)\sim\hat{\mathbb{P}}_n}[(f(x) - y)^2] + \lambda_1\|f\|_\sigma + \lambda_2\|f^2\|_{\sigma/\sqrt{2}}. \tag{22}$$

Given data $(x_i, y_i)_{i=1}^n$, for kernel ridge regression, the Representer Theorem implies that it is sufficient to consider only $f$ of the form $f = \sum_{i=1}^n a_i k_\sigma(x_i, \cdot)$. Here this is not in general possible due to the norm of $f^2$. However, it is possible to evaluate and compute gradients of $\|f^2\|_{\sigma/\sqrt{2}}^2$: let $K$ be the matrix with $K_{ij} = k_{\sqrt{2}\sigma}(x_i, x_j)$, and let $D = \mathrm{diag}(a)$. Using Proposition 4.1, we can prove $\|f^2\|_{\sigma/\sqrt{2}}^2 = \mathrm{tr}((DK)^4)$ A complete proof is given in the appendix.

## 5 Approximation and connections to variance regularization

In the previous section we studied bounding the MMD DRO problem (5) via Hilbert norm penalization. Going beyond kernel methods where we search over $f \in \mathcal{H}$, it is even less clear how to evaluate the Hilbert norm $\|\ell_f\|_\mathcal{H}$. To circumvent this issue, next we approach the DRO problem from a different angle: we directly search for the adversarial distribution $\mathbb{Q}$. Along the way, we will build connections to variance regularization [27, 18, 22, 31], where the empirical risk is regularized by the empirical variance of $\ell_f$: $\mathrm{Var}_{\hat{\mathbb{P}}_n}(\ell_f) = \mathbb{E}_{x\sim\hat{\mathbb{P}}_n}[\ell_f(x)^2] - \mathbb{E}_{x\sim\hat{\mathbb{P}}_n}[\ell_f(x)]^2$. In particular, we show in Theorem 5.1 that MMD DRO yields stronger regularization than variance.

Searching over all distributions $\mathbb{Q}$ in the MMD ball is intractable, so we restrict our attention to those with the same support $\{x_i\}_{i=1}^n$ as the empirical sample $\hat{\mathbb{P}}_n$. All such distributions $\mathbb{Q}$ can be written as $\mathbb{Q} = \sum_{i=1}^n w_i\delta_{x_i}$, where $w$ is in the $n$-dimensional simplex. By restricting the set of candidate distributions $\mathbb{Q}$, we make the adversary weaker:

$$\begin{array}{ll} \sup_{\mathbb{Q}} & \mathbb{E}_{x\sim\mathbb{Q}}[\ell_f(x)] \\ \text{s.t.} & d_{\mathrm{MMD}}(\mathbb{Q}, \hat{\mathbb{P}}_n) \leq \epsilon \end{array} \geq \begin{array}{ll} \sup_w & \sum_{i=1}^n w_i\ell_f(x_i) \\ \text{s.t.} & d_{\mathrm{MMD}}(\sum_{i=1}^n w_i\delta_{x_i}, \hat{\mathbb{P}}_n) \leq \epsilon \\ & \sum_{i=1}^n w_i = 1 \\ & w_i \geq 0\ \forall i = 1,\ldots,n. \end{array} \tag{23}$$

By restricting the support of $\mathbb{Q}$, it is no longer possible to guarantee out of sample performance, since it typically will have different support. Yet, as we will see, problem (23) has nice connections.

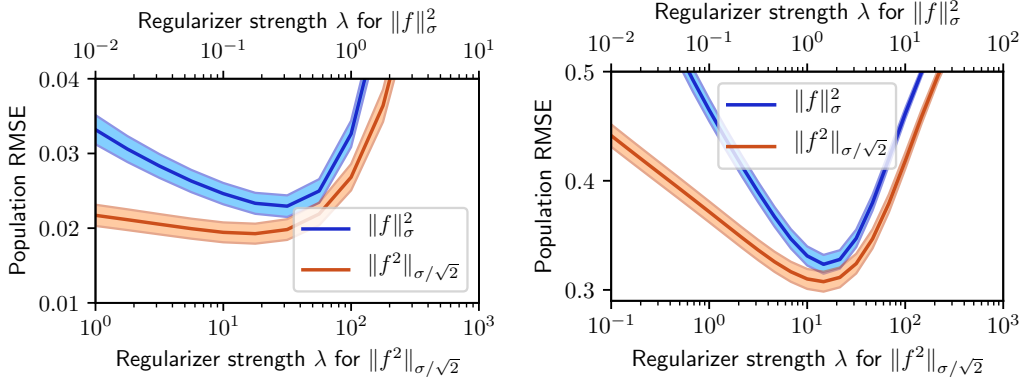

Figure 1: Comparison of the two regularizers $\|f\|_\sigma^2$ and $\|f^2\|_{\sigma/\sqrt{2}}$ in both the easy (left) and hard (right) settings, across a parameter sweep of $\lambda$. The $x$-axis is shifted to make comparison easier.

The $d_{\mathrm{MMD}}$ constraint is a quadratic penalty on $v = w - \frac{1}{n}\mathbf{1}$, as one may see via the mean embedding definition of MMD:

$$d_{\mathrm{MMD}}\left(\sum_{i=1}^n w_i \delta_{x_i}, \hat{\mathbb{P}}_n\right)^2 = \left\|\sum_{i=1}^n w_i k(x_i, \cdot) - \frac{1}{n}\sum_{i=1}^n k(x_i, \cdot)\right\|_{\mathcal{H}}^2 = \left\|\sum_{i=1}^n v_i k(x_i, \cdot)\right\|_{\mathcal{H}}^2. \quad (24)$$

The last term is $v^T K v = (w - \frac{1}{n}\mathbf{1})^T K (w - \frac{1}{n}\mathbf{1})$, where $K$ is the kernel matrix with $K_{ij} = k(x_i, x_j)$. If the radius $\epsilon$ of the uncertainty set is small enough, the constraints $w_i \geq 0$ are inactive, and can be ignored. By dropping these constraints, we can solve the adversary's problem in closed form:

**Lemma 5.1.** *Let $\vec{\ell}$ be the vector with $i$-th element $\ell_f(x_i)$. If $\epsilon$ is small enough that the constraints $w_i$ are not active, then the optimal value of problem* (23) *is given by*

$$\mathbb{E}_{x \sim \hat{\mathbb{P}}_n}[\ell_f(x)] + \epsilon \sqrt{\vec{\ell}^T K^{-1} \vec{\ell} - \frac{(\vec{\ell}^T K^{-1} \mathbf{1})^2}{\mathbf{1}^T K^{-1} \mathbf{1}}}. \quad (25)$$

In other words, fitting a model to minimize the support-constrained approximation of MMD DRO is equivalent to penalizing by the nonconvex regularizer in Lemma 5.1. To better understand this regularizer, consider, for instance, the case that the kernel matrix $K$ equals the identity $I$. This will happen e.g. for a Gaussian kernel as the bandwidth $\sigma$ approaches zero. Then, the regularizer equals

$$\epsilon \sqrt{\vec{\ell}^T K^{-1} \vec{\ell} - \frac{(\vec{\ell}^T K^{-1} \mathbf{1})^2}{\mathbf{1}^T K^{-1} \mathbf{1}}} = \epsilon \sqrt{\vec{\ell}^T \vec{\ell} - \frac{(\vec{\ell}^T \mathbf{1})^2}{\mathbf{1}^T \mathbf{1}}} = \epsilon \sqrt{n} \sqrt{\mathrm{Var}_{\hat{\mathbb{P}}_n}(\ell_f)}. \quad (26)$$

In fact, this equivalence holds a bit more generally:

**Lemma 5.2.** *Let $K = aI + b\mathbf{1}\mathbf{1}^T$. Then,* $\sqrt{\vec{\ell}^T K^{-1} \vec{\ell} - \frac{(\vec{\ell}^T K^{-1} \mathbf{1})^2}{\mathbf{1}^T K^{-1} \mathbf{1}}} = a^{-1/2} \sqrt{n} \sqrt{\mathrm{Var}_{\hat{\mathbb{P}}_n}(\ell_f)}.$

As a consequence, we conclude that with the right choice of kernel $k$, MMD DRO is a stronger regularizer than variance:

**Theorem 5.1.** *There exists a kernel $k$ so that MMD DRO bounds the variance regularized problem:*

$$\mathbb{E}_{x \sim \hat{\mathbb{P}}_n}[\ell_f(x)] \quad \leq \quad \mathbb{E}_{x \sim \hat{\mathbb{P}}_n}[\ell_f(x)] + \epsilon \sqrt{n}\sqrt{\mathrm{Var}_{\hat{\mathbb{P}}_n}(\ell_f)} \quad \leq \quad \sup_{\mathbb{Q}: d_{\mathrm{MMD}}(\mathbb{Q}, \hat{\mathbb{P}}_n) \leq \epsilon}[\ell_f(x)]. \quad (27)$$

## 6 Experiments

In subsection 4.3 we proposed an alternate regularizer for kernel ridge regression, specifically, penalizing $\|f^2\|_{\sigma/\sqrt{2}}$ instead of $\|f\|_\sigma^2$. Here we probe the new regularizer on a synthetic problem where we can precisely compute the population risk $R_{\mathbb{P}}(f)$. Consider the Gaussian kernel $k_\sigma$ with $\sigma = 1$. Fix the ground truth $h = k_\sigma(1, \cdot) - k_\sigma(-1, \cdot) \in \mathcal{H}_\sigma$. Sample $10^4$ points from a standard one dimensional Gaussian, and set this as the population $\mathbb{P}$. Then subsample $n$ points $x_i = h(x_i) + \epsilon_i$,

where $\epsilon_i$ are Gaussian. We consider both an easy regime, where $n = 10^3$ and $\mathrm{Var}(\epsilon_i) = 10^{-2}$, and a hard regime where $n = 10^2$ and $\mathrm{Var}(\epsilon_i) = 1$. On the empirical data, we fit $f \in \mathcal{H}_\sigma$ by minimizing square loss plus either $\lambda \|f\|_\sigma^2$ (as is typical) or $\lambda \|f^2\|_{\sigma/\sqrt{2}}$ (our proposal). We average over $10^2$ resampling trials for the easy case and $10^3$ for the hard case, and report 95% confidence intervals. Figure 1 shows the result in each case for a parameter sweep over $\lambda$. If $\lambda$ is tuned properly, the tighter regularizer $\|f^2\|_{\sigma/\sqrt{2}}$ yields better performance in both cases. It also appears the regularizer $\|f^2\|_{\sigma/\sqrt{2}}$ is less sensitive to the choice of $\lambda$: performance decays slowly when $\lambda$ is too low.

## 7    Conclusion

We introduce MMD DRO, distributionally robust optimization with maximum mean discrepancy uncertainty sets. We prove fundamental structural results and upper bounds for MMD DRO, and unearth deep connections, in particular to Gaussian kernel ridge regression and variance regularization.

Several open questions remain. In terms of theory, our MMD DRO approach to generalization bounds leaves much new ground to explore. In particular, we conjecture that our approach might also work for ridge regression with non-Gaussian kernels. Practically, there is also much left to do to make MMD DRO a general purpose tool. We have presented two approximations of MMD DRO, each with strengths and drawbacks: the upper bound in Corollary 3.2 enables our kernel ridge regression generalization bound, but is potentially loose, and is difficult to use more generally because the Hilbert norm is tricky to compute; the discrete approximation in Section 5 is more practical but is not an upper bound on the MMD DRO problem. Future work could address these drawbacks, or potentially develop a tractable exact reformulation of the DRO problem.

**Acknowledgements**

This work was supported by The Defense Advanced Research Projects Agency (grant number YFA17 N66001-17-1-4039). The views, opinions, and/or findings contained in this article are those of the author and should not be interpreted as representing the official views or policies, either expressed or implied, of the Defense Advanced Research Projects Agency or the Department of Defense. We thank Cameron Musco and Joshua Robinson for helpful conversations, and Marwa El Halabi and Sebastian Claici for comments on the draft.

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
