[Supplementary Material]

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

# A   Proofs of main structural results

*Proof of Theorem 3.1.* We will use weak duality to derive a candidate solution, and then use that solution to show strong duality. First, note that

$$\sup_{\mu_{\mathbb{Q}} \in \mathcal{H}: \|\mu_{\mathbb{Q}} - \mu_{\mathbb{P}}\|_{\mathcal{H}} \leq \epsilon} \langle f, \mu_{\mathbb{Q}} \rangle_{\mathcal{H}} = \sup_{\mu_{\mathbb{Q}} \in \mathcal{H}} \inf_{\lambda \geq 0} \left\{ \langle f, \mu_{\mathbb{Q}} \rangle_{\mathcal{H}} - \lambda(\|\mu_{\mathbb{Q}} - \mu_{\mathbb{P}}\|_{\mathcal{H}}^2 - \epsilon^2) \right\} \tag{28}$$

$$\leq \inf_{\lambda \geq 0} \sup_{\mu_{\mathbb{Q}} \in \mathcal{H}} \left\{ \langle f, \mu_{\mathbb{Q}} \rangle_{\mathcal{H}} - \lambda(\|\mu_{\mathbb{Q}} - \mu_{\mathbb{P}}\|_{\mathcal{H}}^2 - \epsilon^2) \right\} \tag{29}$$

$$= \inf_{\lambda \geq 0} \left\{ \lambda \epsilon^2 + \sup_{\mu_{\mathbb{Q}} \in \mathcal{H}} \left\{ \langle f, \mu_{\mathbb{Q}} \rangle_{\mathcal{H}} - \lambda \|\mu_{\mathbb{Q}} - \mu_{\mathbb{P}}\|_{\mathcal{H}}^2 \right\} \right\}. \tag{30}$$

We first focus on the innermost objective, which may be rewritten:

$$\langle f, \mu_{\mathbb{Q}} \rangle_{\mathcal{H}} - \lambda \|\mu_{\mathbb{Q}} - \mu_{\mathbb{P}}\|_{\mathcal{H}}^2 = \langle f, \mu_{\mathbb{P}} \rangle_{\mathcal{H}} + \langle f, \mu_{\mathbb{Q}} - \mu_{\mathbb{P}} \rangle_{\mathcal{H}} - \lambda \|\mu_{\mathbb{Q}} - \mu_{\mathbb{P}}\|_{\mathcal{H}}^2 \tag{31}$$

$$= \langle f, \mu_{\mathbb{P}} \rangle_{\mathcal{H}} - \lambda \left[ \|\mu_{\mathbb{Q}} - \mu_{\mathbb{P}}\|_{\mathcal{H}}^2 - 2 \left\langle \frac{1}{2\lambda} f, \mu_{\mathbb{Q}} - \mu_{\mathbb{P}} \right\rangle_{\mathcal{H}} \right] \tag{32}$$

$$= \langle f, \mu_{\mathbb{P}} \rangle_{\mathcal{H}} - \lambda \left[ \left\| \mu_{\mathbb{Q}} - \mu_{\mathbb{P}} - \frac{1}{2\lambda} f \right\|_{\mathcal{H}}^2 + \left\| \frac{1}{2\lambda} f \right\|_{\mathcal{H}}^2 \right], \tag{33}$$

where the final inequality is by completing the square. Only one term depends on $\mu_{\mathbb{Q}}$, namely $-\lambda \|\mu_{\mathbb{Q}} - \mu_{\mathbb{P}} - \frac{1}{2\lambda} f\|_{\mathcal{H}}^2$; since norms are nonnegative, this term can never exceed zero, and zero is achieved by $\mu_{\mathbb{Q}}^* = \mu_{\mathbb{P}} + \frac{1}{2\lambda} f \in \mathcal{H}$, yielding inner objective value

$$\langle f, \mu_{\mathbb{P}} \rangle_{\mathcal{H}} - \lambda \left\| \frac{1}{2\lambda} f \right\|_{\mathcal{H}}^2 = \langle f, \mu_{\mathbb{P}} \rangle_{\mathcal{H}} - \frac{1}{4\lambda} \|f\|_{\mathcal{H}}^2. \tag{34}$$

Plugging this in for the inner problem, and then solving for the optimal dual variable $\lambda^*$, we derive the upper bound:

$$\sup_{\mu_{\mathbb{Q}} \in \mathcal{H}: \|\mu_{\mathbb{Q}} - \mu_{\mathbb{P}}\|_{\mathcal{H}} \leq \epsilon} \langle f, \mu_{\mathbb{Q}} \rangle_{\mathcal{H}} \leq \inf_{\lambda \geq 0} \left\{ \lambda \epsilon^2 + \langle f, \mu_{\mathbb{P}} \rangle_{\mathcal{H}} + \frac{1}{4\lambda} \|f\|_{\mathcal{H}}^2 \right\} \tag{35}$$

$$= \langle f, \mu_{\mathbb{P}} \rangle_{\mathcal{H}} + \epsilon \|f\|_{\mathcal{H}}. \tag{36}$$

The optimal dual variable $\lambda^* = \frac{1}{2\epsilon} \|f\|_{\mathcal{H}}$ is that which balances the two terms. Plugging this in, we find that $\mu_{\mathbb{Q}}^* = \mu_{\mathbb{P}} + \frac{\epsilon}{\|f\|_{\mathcal{H}}} f$.

In order to prove equality, it remains to show strong duality holds. We will achieve this by lower bounding the original objective. Specifically, the supremum over all $\mu_{\mathbb{Q}}$ can be lower bounded by plugging in our particular $\mu_{\mathbb{Q}}^*$:

$$\sup_{\mu_{\mathbb{Q}} \in \mathcal{H}: \|\mu_{\mathbb{Q}} - \mu_{\mathbb{P}}\|_{\mathcal{H}} \leq \epsilon} \langle f, \mu_{\mathbb{Q}} \rangle_{\mathcal{H}} = \sup_{\mu_{\mathbb{Q}} \in \mathcal{H}} \inf_{\lambda \geq 0} \left\{ \langle f, \mu_{\mathbb{Q}} \rangle_{\mathcal{H}} - \lambda(\|\mu_{\mathbb{Q}} - \mu_{\mathbb{P}}\|_{\mathcal{H}}^2 - \epsilon^2) \right\} \tag{37}$$

$$\geq \inf_{\lambda \geq 0} \left\{ \langle f, \mu_{\mathbb{Q}}^* \rangle_{\mathcal{H}} - \lambda(\|\mu_{\mathbb{Q}}^* - \mu_{\mathbb{P}}\|_{\mathcal{H}}^2 - \epsilon^2) \right\} \tag{38}$$

$$= \inf_{\lambda \geq 0} \left\{ \left\langle f, \mu_{\mathbb{P}} + \frac{\epsilon}{\|f\|_{\mathcal{H}}} f \right\rangle_{\mathcal{H}} - \lambda \left( \left\| \frac{\epsilon}{\|f\|_{\mathcal{H}}} f \right\|_{\mathcal{H}}^2 - \epsilon^2 \right) \right\} \tag{39}$$

$$= \inf_{\lambda \geq 0} \left\{ \left\langle f, \mu_{\mathbb{P}} + \frac{\epsilon}{\|f\|_{\mathcal{H}}} f \right\rangle_{\mathcal{H}} - \lambda \left( \epsilon^2 - \epsilon^2 \right) \right\} \tag{40}$$

$$= \left\langle f, \mu_{\mathbb{P}} + \frac{\epsilon}{\|f\|_{\mathcal{H}}} f \right\rangle_{\mathcal{H}} = \langle f, \mu_{\mathbb{P}} \rangle_{\mathcal{H}} + \epsilon \|f\|_{\mathcal{H}}. \tag{41}$$

Since the same bound appears on both sides, we have equality.   $\square$

# B   Gaussian kernel bounds

We first reproduce Proposition 4.1 for convenience:

**Proposition B.1.** *Let $f, g \in \mathcal{H}_\sigma$ have the expansions $f = \sum_i a_i k_\sigma(x_i, \cdot)$ and $g = \sum_j b_j k_\sigma(x_j, \cdot)$. For shorthand denote by $z_i = \phi_{\sqrt{2}\sigma}(x_i)$ the (possibly infinite) feature expansion of $x_i$ in $\mathcal{H}_{\sqrt{2}\sigma}$. Then*

$$\|fg\|^2_{\sigma/\sqrt{2}} = \mathrm{tr}(A^2 B^2), \quad \|f\|^2_\sigma = \mathrm{tr}(A^2), \quad \text{and} \quad \|g\|^2_\sigma = \mathrm{tr}(B^2),$$

*where $A = \sum_i a_i z_i z_i^T$ and $B = \sum_j a_j z_j z_j^T$.*

In order to prove Proposition 4.1, we will need a utility lemma that helps translate between $\mathcal{H}_\sigma$ and $\mathcal{H}_{\sigma/\sqrt{2}}$:

**Lemma B.1.** *Let $\langle \cdot, \cdot \rangle_{\sigma/\sqrt{2}}$ be the inner product in the RKHS $\mathcal{H}_{\sigma/\sqrt{2}}$. Let $\langle \cdot, \cdot \rangle_{\sigma'}$ refer to the inner product in $H_{\sigma'}$. Then,*

$$\langle k_\sigma(x, \cdot)k_\sigma(y, \cdot), k_\sigma(a, \cdot)k_\sigma(b, \cdot) \rangle_{\sigma/\sqrt{2}} \tag{42}$$

*can be simplified as*

$$k_{\sigma\sqrt{2}}(x, a)k_{\sigma\sqrt{2}}(x, b)k_{\sigma\sqrt{2}}(y, a)k_{\sigma\sqrt{2}}(y, b). \tag{43}$$

In order to make the proof cleaner, we first derive a couple of identities involving norms and sums.

**Lemma B.2.** *For vectors $x, y, z$, the following identity holds:*

$$\|x - z\|^2 + \|y - z\|^2 = \frac{1}{2}\|x - y\|^2 + 2\left\|z - \frac{x+y}{2}\right\|^2 \tag{44}$$

*Proof.* Simply expand:

$$\|x - z\|^2 + \|y - z\|^2 = \|x\|^2 + \|y\|^2 + 2\|z\|^2 - 2z^T(x + y) \tag{45}$$

$$= \|x\|^2 + \|y\|^2 + 2\left(\|z\|^2 - 2z^T\left(\frac{x+y}{2}\right)\right) \tag{46}$$

$$= \|x\|^2 + \|y\|^2 + 2\left(\left\|z - \frac{x+y}{2}\right\|^2 - \left\|\frac{x+y}{2}\right\|^2\right) \tag{47}$$

$$= \|x\|^2 + \|y\|^2 + 2\left\|z - \frac{x+y}{2}\right\|^2 - \frac{1}{2}\|x + y\|^2 \tag{48}$$

$$= \frac{1}{2}\|x - y\|^2 + 2\left\|z - \frac{x+y}{2}\right\|^2. \qquad \square$$

**Lemma B.3.** *Let $x, y, a, b$ be arbitrary vectors, and define $S$ and $T$ by:*

$$S := \|x - y\|^2 + \|a - b\|^2 + \|(x + y) - (a + b)\|^2$$
$$T := \|x - a\|^2 + \|x - b\|^2 + \|y - a\|^2 + \|y - b\|^2.$$

*Then $S = T$.*

*Proof.* Start by expanding the third term of $S$:

$$\|x - y\|^2 + \|a - b\|^2 + \|(x + y) - (a + b)\|^2 \tag{49}$$
$$= \|x - y\|^2 + \|a - b\|^2 + \|(x - a) + (y - b)\|^2 \tag{50}$$
$$= \|x - y\|^2 + \|a - b\|^2 + 2(x - a)^T(y - b) + \|x - a\|^2 + \|y - b\|^2. \tag{51}$$

The first three terms of equation (51) can be expanded as

$$\|x - y\|^2 + \|a - b\|^2 + 2(x - a)^T(y - b) \tag{52}$$
$$= \|x\|^2 + \|y\|^2 - 2x^T y + \|a\|^2 + \|b\|^2 - 2a^T b + 2(x - a)^T(y - b) \tag{53}$$
$$= \|x\|^2 + \|y\|^2 - 2x^T y + \|a\|^2 + \|b\|^2 - 2a^T b + 2x^T y - 2x^T b - 2a^T y + 2a^T b \tag{54}$$
$$= \|x\|^2 + \|y\|^2 + \|a\|^2 + \|b\|^2 - 2x^T b - 2a^T y \tag{55}$$
$$= \|x - b\|^2 + \|y - a\|^2. \tag{56}$$

Replacing the first three terms in equation (51) by $\|x - b\|^2 + \|y - a\|^2$ yields $T$, i.e. $S = T$. $\square$

We are now equipped to prove Lemma B.1:

*Proof of Lemma B.1.* First, write

$$k_\sigma(x, z)k_\sigma(y, z) = \exp\left(-\frac{1}{2\sigma^2}\left(\|x - z\|^2 + \|y - z\|^2\right)\right) \tag{57}$$

$$= \exp\left(-\frac{1}{2\sigma^2}\left(\frac{1}{2}\|x - y\|^2 + 2\left\|z - \frac{x + y}{2}\right\|^2\right)\right) \tag{58}$$

$$= \exp\left(-\frac{1}{4\sigma^2}\|x - y\|^2\right)\exp\left(-\frac{1}{\sigma^2}\left\|z - \frac{x + y}{2}\right\|^2\right) \tag{59}$$

$$= k_{\sigma\sqrt{2}}(x, y)k_{\sigma/\sqrt{2}}\left(z, \frac{x + y}{2}\right), \tag{60}$$

where in the second line we used Lemma B.2. Note that the first term does not depend on $z$. Now, applying this identity to Equation (42), we find:

$$\langle k_\sigma(x, \cdot)k_\sigma(y, \cdot), k_\sigma(a, \cdot)k_\sigma(b, \cdot)\rangle_{\sigma/\sqrt{2}} \tag{61}$$

$$= k_{\sigma\sqrt{2}}(x, y)k_{\sigma\sqrt{2}}(a, b)\left\langle k_{\sigma/\sqrt{2}}\left(\frac{x + y}{2}, \cdot\right), k_{\sigma/\sqrt{2}}\left(\frac{a + b}{2}, \cdot\right)\right\rangle_{\sigma/\sqrt{2}} \tag{62}$$

$$= k_{\sigma\sqrt{2}}(x, y)k_{\sigma\sqrt{2}}(a, b)k_{\sigma/\sqrt{2}}\left(\frac{x + y}{2}, \frac{a + b}{2}\right) \tag{63}$$

$$= k_{\sigma\sqrt{2}}(x, y)k_{\sigma\sqrt{2}}(a, b)k_{\sigma\sqrt{2}}\left(x + y, a + b\right). \tag{64}$$

To simplify this expression, notice that it takes the form $\exp(-S/(4\sigma^2))$, where

$$S = \|x - y\|^2 + \|a - b\|^2 + \|(x + y) - (a + b)\|^2. \tag{65}$$

By Lemma B.3, $S$ is equal to

$$S = \|x - a\|^2 + \|x - b\|^2 + \|y - a\|^2 + \|y - b\|^2, \tag{66}$$

which means equation (64) can be rewritten as

$$\exp\left(-\frac{S}{4\sigma^2}\right) = \exp\left(-\frac{\|x - a\|^2}{4\sigma^2}\right)\exp\left(-\frac{\|x - b\|^2}{4\sigma^2}\right)\exp\left(-\frac{\|y - a\|^2}{4\sigma^2}\right)\exp\left(-\frac{\|y - b\|^2}{4\sigma^2}\right)$$

$$= k_{\sigma\sqrt{2}}(x, a)k_{\sigma\sqrt{2}}(x, b)k_{\sigma\sqrt{2}}(y, a)k_{\sigma\sqrt{2}}(y, b).$$

$$\square$$

With Lemma B.1 available, it is possible to prove Proposition 4.1:

*Proof of Proposition 4.1.* Define the vectors $z_i$ as described, so that $z_i^T z_j = k_{\sqrt{2}\sigma}(x_i, x_j)$. For convenience, also write $K_{ij} = k_{\sqrt{2}\sigma}(x_i, x_j)$, and observe that $K_{ij}^2 = k_\sigma(x_i, x_j)$. It follows that

$$\|f\|_\sigma^2 = \sum_i \sum_j a_i a_j k_\sigma(x_i, x_j) = \sum_i \sum_j a_i a_j K_{ij}^2 = \sum_i \sum_j a_i a_j z_i^T z_j z_j^T z_i \tag{67}$$

Rearranging the inner terms, we find

$$\|f\|_\sigma^2 = \sum_i a_i z_i^T \left(\sum_j a_j z_j z_j^T\right) z_i = \sum_i a_i z_i^T A z_i = \text{tr}\left(\sum_i a_i z_i z_i^T A\right) = \text{tr}(A^2), \tag{68}$$

where we have used the definition of $A$, the fact that the trace of a scalar is simply that scalar, and the cyclic property of the trace. The proof that $\|g\|_\sigma^2 = \text{tr}(B^2)$ is identical, so we omit it.

The derivation of the trace form of $\|fg\|_{\sigma/\sqrt{2}}^2$ is more complicated. Expanding out $fg$, we see that

$$(fg)(x) = \sum_{i,j} a_i b_j k_\sigma(x_i, x)k_\sigma(x_j, x). \tag{69}$$

Therefore the norm $\|fg\|^2_{\sigma/\sqrt{2}}$, which is simply $\langle fg, fg \rangle_{\sigma/\sqrt{2}}$, is equal to:

$$\langle fg, fg \rangle_{\sigma/\sqrt{2}} = \left\langle \sum_{i,j} a_i b_j k_\sigma(x_i, x) k_\sigma(x_j, x), \sum_{i',j'} a_{i'} b_{j'} k_\sigma(x_{i'}, x) k_\sigma(x_{j'}, x) \right\rangle_{\sigma/\sqrt{2}} \tag{70}$$

$$= \sum_{i,j,i',j'} a_i a_{i'} b_j b_{j'} \langle k_\sigma(x_i, x) k_\sigma(x_j, x), k_\sigma(x_{i'}, x) k_\sigma(x_{j'}, x) \rangle_{\sigma/\sqrt{2}} \tag{71}$$

$$= \sum_{i,j,i',j'} a_i a_{i'} b_j b_{j'} k_{\sigma\sqrt{2}}(x_i, x_{i'}) k_{\sigma\sqrt{2}}(x_i, x_{j'}) k_{\sigma\sqrt{2}}(x_j, x_{i'}) k_{\sigma\sqrt{2}}(x_j, x_{j'}) \tag{72}$$

$$= \sum_{i,j,i',j'} a_i a_{i'} b_j b_{j'} K_{ii'} K_{ij'} K_{ji'} K_{jj'}, \tag{73}$$

where in the second to last step we have used Lemma B.1. Before continuing, observe the identity

$$\sum_\ell a_\ell K_{i\ell} K_{j\ell} = \sum_\ell a_\ell z_i^T z_\ell z_\ell^T z_j = z_i^T \left( \sum_\ell a_\ell z_\ell z_\ell^T \right) z_j = z_i^T A z_j \tag{74}$$

Similarly, $\sum_\ell b_\ell K_{i\ell} K_{j\ell} = z_i^T B z_j$. Leveraging these identities, we continue:

$$\sum_{i,j,i',j'} a_i a_{i'} b_j b_{j'} K_{ii'} K_{ij'} K_{ji'} K_{jj'} = \sum_{i,i',j} a_i a_{i'} b_j K_{ii'} K_{ji'} \sum_{j'} b_{j'} K_{ij'} K_{jj'} \tag{75}$$

$$= \sum_{i,i',j} a_i a_{i'} b_j K_{ii'} K_{ji'} (z_i^T B z_j) \tag{76}$$

$$= \sum_{i,j} a_i b_j \left( \sum_{i'} a_{i'} K_{ii'} K_{ji'} \right) (z_i^T B z_j) \tag{77}$$

$$= \sum_{i,j} a_i b_j (z_j^T A z_i)(z_i^T B z_j). \tag{78}$$

At this point we leverage the cyclic property of the trace, so the above expression equals:

$$\mathrm{tr}\left( \sum_{i,j} a_i b_j A z_i z_i^T B z_j z_j^T \right) = \mathrm{tr}\left( A \left( \sum_i a_i z_i z_i^T \right) B \left( \sum_j b_j z_j z_j^T \right) \right) = \mathrm{tr}(A^2 B^2). \quad \square$$

## B.1 Trace inequality

*Proof of Lemma 4.1.* Consider the trace inner product $\langle X, Y \rangle = \mathrm{tr}(X^T Y) = \mathrm{tr}(XY)$, where the final equality is because $X$ is symmetric. By the Cauchy-Schwarz inequality, we have $\mathrm{tr}(XY) \leq \sqrt{\mathrm{tr}(X^2)\mathrm{tr}(Y^2)}$. Let $\{\lambda_i\}_{i=1}^n$ be the eigenvalues of $X$. Then,

$$\mathrm{tr}(X^2) = \sum_{i=1}^n \lambda_i^2 \leq \sum_{i=1}^n \lambda_i^2 + 2\sum_{i=1}^n \sum_{j=i+1}^n \lambda_i \lambda_j = \left( \sum_{i=1}^n \lambda_i \right)^2 = \mathrm{tr}(X)^2, \tag{79}$$

where the inequality holds because $\lambda_i$ are all nonnegative. The same holds for any positive semidefinite matrix, in particular, $Y$. Combining these two inequalities, we have

$$\mathrm{tr}(XY) \leq \sqrt{\mathrm{tr}(X^2)\mathrm{tr}(Y^2)} \leq \sqrt{\mathrm{tr}(X)^2 \mathrm{tr}(Y)^2} = \mathrm{tr}(X)\mathrm{tr}(Y). \tag{80}$$

$\square$

## B.2 Extensions of Proposition 4.1

There are many useful corollaries and extensions of Proposition 4.1. First, we give a result that makes it tractable to actually compute $\|fg\|_{\sigma/\sqrt{2}}$:

**Corollary B.1.** *Suppose $f = \sum_{i=1}^{n} a_i k_\sigma(x_i, \cdot)$ and $g = \sum_{i=1}^{n} b_i k_\sigma(x_i, \cdot)$ have the same finite expansion, but with potentially different coefficients. Form the kernel matrix $K$ with $K_{ij} = k_{\sqrt{2}\sigma}(x_i, x_j)$, where we have replaced the bandwidth $\sigma$ with $\sqrt{2}\sigma$. Write $D_a = \mathrm{diag}(a)$ and similarly for $D_b$. Then,*

$$\|fg\|^2_{\sigma/\sqrt{2}} = \mathrm{tr}((D_a K)^2 (D_b K)^2). \tag{81}$$

*Proof.* Pick vectors $z_i$ so that $z_i^T z_j = K_{ij}$, and let $Z$ be the matrix with $i$-th column $z_i$. Note that $A = \sum_{i=1}^{n} a_i z_i z_i^T = Z D_a Z^T$, and similarly for $B$. Then we may write

$$\|fg\|^2_{\sigma/\sqrt{2}} \overset{(a)}{=} \mathrm{tr}(A^2 B^2) \tag{82}$$

$$= \mathrm{tr}((Z D_a Z^T)(Z D_a Z^T)(Z D_b Z^T)(Z D_b Z^T)) \tag{83}$$

$$\overset{(b)}{=} \mathrm{tr}(D_a Z^T Z D_a Z^T Z D_b Z^T Z D_b Z^T Z) \tag{84}$$

$$\overset{(c)}{=} \mathrm{tr}(D_a K D_a K D_b K D_b K) \tag{85}$$

$$= \mathrm{tr}((D_a K)^2 (D_b K)^2), \tag{86}$$

where (a) is by Proposition 4.1, (b) is by the cyclic property of the trace, and (c) follows since $Z^T Z = K$ by definition of $z_i$. $\qquad\square$

## C  Proofs for Section 5

*Proof of Lemma 5.1.* For notational convenience, we just write $\ell$ instead of $\vec{\ell}$. First, notice that problem (23), once the $w_i \geq 0$ constraint is dropped, can be written

$$
\begin{aligned}
\sup_w \quad & \ell^T w \\
\text{s.t.} \quad & \left(w - \tfrac{1}{n}\mathbf{1}\right)^T K \left(w - \tfrac{1}{n}\mathbf{1}\right) \leq \epsilon^2 \\
& \mathbf{1}^T w = 1
\end{aligned}
\tag{87}
$$

Write $v = w - \frac{1}{n}\mathbf{1}$. Then the value of problem (87) is equal to

$$
\frac{1}{n}\mathbf{1}^T \ell + 
\begin{aligned}
\sup_v \quad & \ell^T v \\
\text{s.t.} \quad & v^T K v \leq \epsilon^2 \\
& \mathbf{1}^T v = 0
\end{aligned}
\tag{88}
$$

and we can focus on this slightly simpler problem. This problem can be in turn rewritten as:

$$\sup_v \inf_{\eta \geq 0, \lambda} \left\{ \ell^T v - \eta(v^T K v - \epsilon^2) - \lambda \mathbf{1}^T v \right\}. \tag{89}$$

By Slater's condition, strong duality holds, so the optimal value is equal to:

$$\inf_{\eta \geq 0, \lambda} \left\{ \eta\epsilon^2 + \sup_v \left\{ \ell^T v - \eta v^T K v - \lambda \mathbf{1}^T v \right\} \right\} \tag{90}$$

$$= \inf_{\eta \geq 0, \lambda} \left\{ \eta\epsilon^2 + \sup_v \left\{ -\eta v^T K v + (\ell - \lambda\mathbf{1})^T v \right\} \right\}. \tag{91}$$

The inner problem is a concave quadratic maximization problem. In general, if $A$ is symmetric, $-x^T A x + b^T x$ is maximized when $x = \frac{1}{2} A^{-1} b$, and the resulting objective value is $\frac{1}{4} b^T A^{-1} b$. Applying this to the problem at hand, we find that the optimal $v^*$ satisfies:

$$v^* = \frac{1}{2\eta} K^{-1}(\ell - \lambda\mathbf{1}), \tag{92}$$

and the corresponding objective value of the inner problem is

$$\frac{1}{4\eta}(\ell - \lambda\mathbf{1})^T K^{-1}(\ell - \lambda\mathbf{1}). \tag{93}$$

The overall problem is therefore

$$\inf_{\eta \geq 0, \lambda} \left\{ \eta \epsilon^2 + \frac{1}{4\eta} (\ell - \lambda \mathbf{1})^T K^{-1} (\ell - \lambda \mathbf{1}) \right\}. \tag{94}$$

The objective is a convex quadratic in $\lambda$, and it is simple to check that $\lambda^* = (\mathbf{1}^T K^{-1} \ell)/(\mathbf{1}^T K^{-1} \mathbf{1})$. Then, both remaining terms are positive, so it is optimal to balance them. This leads to

$$\eta^* \epsilon^2 = \frac{1}{4\eta^*} (\ell - \lambda^* \mathbf{1})^T K^{-1} (\ell - \lambda^* \mathbf{1}) \tag{95}$$

$$\implies \frac{1}{2\eta^*} = \frac{\epsilon}{\sqrt{(\ell - \lambda^* \mathbf{1})^T K^{-1} (\ell - \lambda^* \mathbf{1})}}, \tag{96}$$

and the overall optimal value is

$$2 \cdot \frac{1}{4\eta^*} (\ell - \lambda^* \mathbf{1})^T K^{-1} (\ell - \lambda^* \mathbf{1}) \tag{97}$$

$$= \epsilon \sqrt{(\ell - \lambda^* \mathbf{1})^T K^{-1} (\ell - \lambda^* \mathbf{1})}. \tag{98}$$

The term inside the square root is equal to

$$(\ell - \lambda^* \mathbf{1})^T K^{-1} (\ell - \lambda^* \mathbf{1}) = \ell^T K^{-1} \ell - 2\lambda^* \mathbf{1}^T K^{-1} \ell + (\lambda^*)^2 \mathbf{1}^T K^{-1} \mathbf{1} \tag{99}$$

$$= \ell^T K^{-1} \ell - \frac{(\mathbf{1}^T K^{-1} \ell)^2}{\mathbf{1}^T K^{-1} \mathbf{1}}, \tag{100}$$

from which we can simply compute the overall objective of the original problem. $\qquad \square$

*Proof of Lemma 5.2.* One can prove via the matrix inversion lemma that

$$K^{-1} = (aI + b\mathbf{1}\mathbf{1}^T)^{-1} = a^{-1} \left[ I - \frac{b}{a + bn} \mathbf{1}\mathbf{1}^T \right]. \tag{101}$$

As a consequence,

$$a\ell^T K^{-1} \ell = \|\ell\|^2 - \frac{b}{a + bn} (\mathbf{1}^T \ell)^2 \tag{102}$$

$$a\ell^T K^{-1} \mathbf{1} = \mathbf{1}^T \ell - \frac{b}{a + bn} (\mathbf{1}^T \ell)(\mathbf{1}^T \mathbf{1}) = \frac{a}{a + bn} \cdot \mathbf{1}^T \ell \tag{103}$$

$$a\mathbf{1}^T K^{-1} \mathbf{1} = \mathbf{1}^T \mathbf{1} - \frac{b}{a + bn} (\mathbf{1}^T \mathbf{1})^2 = \frac{a}{a + bn} \cdot n. \tag{104}$$

It follows that

$$\frac{(a\ell^T K^{-1} \mathbf{1})^2}{a\mathbf{1}^T K^{-1} \mathbf{1}} = a \cdot \frac{\left(\frac{a}{a+bn} \mathbf{1}^T \ell\right)^2}{\frac{a}{a+bn} \cdot n} = a \cdot \frac{(\mathbf{1}^T \ell)^2}{n} \cdot \frac{a}{a + bn} = a \cdot (\mathbf{1}^T \ell)^2 \cdot \left( \frac{1}{n} - \frac{b}{a + bn} \right) \tag{105}$$

and therefore

$$a \cdot \left[ \ell^T K^{-1} \ell - \frac{(\ell^T K^{-1} \mathbf{1})^2}{\mathbf{1}^T K^{-1} \mathbf{1}} \right] = a \cdot \left[ \|\ell\|^2 - \frac{b}{a + bn} (\mathbf{1}^T \ell)^2 - (\mathbf{1}^T \ell)^2 \cdot \left( \frac{1}{n} - \frac{b}{a + bn} \right) \right] \tag{106}$$

$$= a \cdot \left[ \|\ell\|^2 - \frac{(\mathbf{1}^T \ell)^2}{n} \right] = a \cdot \mathrm{Var}_{\hat{\mathbb{P}}_n}(\ell), \tag{107}$$

from which the conclusion follows. $\qquad \square$