[Reviews · NeurIPS 2019]

Reviewer 1



=== after author's rebuttal == The study of MMD DRO in the submission seems novel, and I believe understanding between MMD DRO and ML are important both theoretically and practically. I raised my score from 4 to 6 after reading the author's feedback, mainly due to the novelty of the framework. However, I would expect the author can provide a thorough discussion of the limitation of the result in the camera-ready version. Weakness: Due to the intractbility of the MMD DRO problem, the submission did not find an exact reformulation as much other literature in DRO did for other probability metrics. Instead, the author provides several layers of approximation. The reason why I emphasize the importance of a tight bound, if not an exact reformulation, is that one of the major criticism about (distributionally) robust optimization is that it is sometimes too conservative, and thus a loose upper bound might not be sufficient to mitigate the over-conservativeness and demonstrate the power of distributionally robust optimization. When a new distance is introduced into the DRO framework, a natural question is why it should be used compared with other existing approaches. I hope there will be a more fair comparision in the camera-ready version. =============== 1. The study of MMD DRO is mostly motivated by the poor out-of-sample performance of existing phi-divergence and Wasserstein uncertainty sets. However, I don't believe this is indeed the case. For example, Namkoong and Duchi (2016), and Blanchet, Kang, and Murthy (2016) show the dimension-independent bound 1/\sqrt{n} for a broad class of objective functions in the case of phi-divergence and Wasserstein metric respectively. They didn't require the population distribution to be within the uncertainty set, and in fact, such a requirement is way too conservative and it is exactly what they wanted to avoid. 2. Unlike phi-divergence or Wasserstein uncertainty sets, MMD DRO seems not enjoy a tractable exact equivalent reformulation, which seems to be a severe drawback to me. The upper bound provided in Theorem 3.1 is crude especially because it drops the nonnegative constraint on the distribution, and further approximation is still needed even applied to a simple kernel ridge regression problem. Moreover, it seems restrictive to assume the loss \ell_f belongs to the RKHS as already pointed out by the authors. 3. I am confused about the statement in Theorem 5.1, as it might indicate some disadvantage of MMD DRO, as it provides a more conservative upper bound than the variance regularized problem. 4. Given the intractability of the MMD DRO and several layers of approximation, the numerical experiment in Section 6 is insufficient to demonstrate the usefulness of the new framework. References: Namkoong, H. and Duchi, J.C., 2017. Variance-based regularization with convex objectives. In Advances in Neural Information Processing Systems (pp. 2971-2980). Blanchet, J., Kang, Y. and Murthy, K., 2016. Robust wasserstein profile inference and applications to machine learning. arXiv preprint arXiv:1610.05627.

Reviewer 2



I enjoyed reading the paper so much. It was well written. As the author states, the paper presents four original contributions. MMD DRO, its generalization bound, its tractable approximation, and new regularizer for kernel ridge regression. They are all indeed novel and beyond just a combination of existing techniques. The paper would be a food for thought for many researchers. A concern I have is that the approximation of the bound for a general learning problem might lead to a population distribution not included in the uncertain set. Therefore, to my understanding, the proposed approximation shares the weak point of phi-divergence approach. If so, shouldn't it be stated more clearly? --------------------------after rebuttal----------------------------------------------------- The authors claim that the MMD DRO solves the problem of the uncertainty set, e.g. line 40. However, at this point it is not practical since the tractable exact reformulation is not presented. Instead only the discrete approximation of the MMD DRO is introduced in Section 5, which might lead to a population distribution not included in the uncertain set. So I personally feel that the authors exaggerated their contribution in the Introduction slightly. I recommend the authors to discuss the limitation of the paper more candidly in the camera ready.

Reviewer 3



Originality: I'm not extremely familiar with the DRO literature, but, as far as I know, the use of MMD for DRO, as well as the connection to kernel ridge regression (KRR), and the proposed regularizer are novel. The generalization bound for KRR is not novel, but the proof approach is novel, and could likely be applied to other problems. Quality: The proofs in the main text appear correct. The experiments are brief but illustrative. Some more discussion of future directions of this work would be nice (the paper ends quite abruptly). Clarity: The majority of this paper is very clearly written. However, I have some concerns about how the distinctions between MMD and Wasserstein DRO are presented (see point 1) under "Improvements"). The provided code is easy to run and follow. Significance: As far as I know DRO in general has received far less attention in the Stat/ML community than in, say, OR and other areas, despite being highly relevant to ML problems. Given that this paper not only provides a clear presentation of the DRO framework for a Stat/ML audience but also supplies novel implications for a standard learning algorithm (kernel ridge regression), I think this paper would be a useful contribution to NeurIPS. -------------------------AFTER REBUTTAL/DISCUSSION------------------------- I thank the reviewers for their clear responses. I continue to feel that the paper can be useful to many members of the NeurIPS community, through its very clear exposition of DRO and the explicit connections it provides between DRO and learning theory. Hence, I continue to support accepting the paper. Reviewer #2, who is most familiar with the DRO literature, has pointed that some of the paper's technical aspects are perhaps not as strong as previous work in DRO rather than providing an exact computationally tractable reformulation, approximations are used, and the bounds are potentially loose. The resulting DRO method may be overly conservative. If the paper is accepted, I would strongly urge the authors to add a conclusion section that discusses these limitations and the potential need for future work along these lines.

[Author Response · NeurIPS 2019]

We thank the reviewers for their helpful comments and enthusiasm.

**Reviewer #2:**   Thank you for your comments. Regarding your points:

**1. Motivation:** We would like to clarify our aims: our main goal is to study the connections and properties of MMD DRO, to reveal its potential benefits, and aid a better general understanding of the DRO landscape with different divergence measures.

We do not claim that Wasserstein or phi-divergence uncertainty are bad or yield poor out-of-sample bounds; we merely highlight differences between them and MMD DRO. For example, in discussing phi-divergences, we state in line 88 that one cannot obtain a generalization bound via Principle 2.1 alone (to be fair, we will add a reminder to the reader that e.g. Namkoong and Duchi (2017) achieve generalization bounds via other means). For Wasserstein, we focus in lines 34-36 and lines 89-99 on other complications, such as difficulty of optimization, and that many of the upper bounds are only asymptotic. Indeed, the convergence results of Blanchet, Kang, and Murthy (2016) (in Section 3 of their paper) are also asymptotic in nature.

**2. Upper bounds instead of exact reformulation, and assuming $\ell_f$ is in an RKHS:** We again emphasize that there are tradeoffs between all three DRO approaches. While discarding the non-negativity constraint in MMD DRO may weaken the bounds, in return we obtain a simple non-asymptotic upper bound (unlike Wasserstein).

Moreover, we contend that the assumption that $\ell_f$ is in an RKHS $\mathcal{H}$ is not so restrictive after all. If the kernel $k$ is universal, as is the case for many kernels used in practice such as Gaussian and Laplace kernels, we can readily extend our results to all bounded continuous functions as described below. We will add this clarification to the paper:

Suppose the loss $\ell_f$ of our predictor $f$ is any bounded continuous function on a compact metric space $\mathcal{X}$. By definition [Muandet et al., Definition 3.3] if $k$ is a universal kernel on $\mathcal{X}$ (associated with the RKHS $\mathcal{H}$), then for any $\epsilon > 0$, there is some $\ell' \in \mathcal{H}$ with $\sup_{x \in \mathcal{X}} |\ell_f(x) - \ell'(x)| < \epsilon$. It follows that for any measure $\mathbb{P}$, we can bound the expectation of $\ell_f(x)$ by that of $\ell'$: $\mathbb{E}_{x \sim \mathbb{P}}[\ell_f(x)] < \mathbb{E}_{x \sim \mathbb{P}}[\ell'(x)] + \epsilon$. Then, we can apply our results to $\ell' \in \mathcal{H}$.

**3. MMD DRO is a more conservative upper bound (Theorem 5.1):** Separate from the task of producing a valid (and hopefully tight) upper bound is the task of designing a regularizer that is practically useful. And stronger regularizers are often better. One drawback of the previously considered chi-squared DRO/variance regularization is that the regularization ceases to have any effect when the training data can be fit perfectly e.g. in deep learning (since then the loss for each datapoint is zero, and so the variance of the loss on the dataset is also zero). In such a regime, stronger penalties such as the RKHS norm continue to be meaningful.

**Reviewer #3:**   Thank you for your support.

**Re: discrete approximation of MMD DRO uncertain set may not contain the population:** Yes, any such discrete approximation can have similar issues. We present it mainly to link variance regularization and MMD DRO, e.g. as in Theorem 5.1.

**Reviewer #4:**   Thank you for your feedback and support.

Before addressing your main comments in detail, we emphasize that at a high level we hope to present MMD DRO as an alternative worth studying, with complementary properties to existing techniques and rich connections. In that context,

**(a) Wasserstein convergence with fewer assumptions:** Thank you for pointing us to the references on non-Euclidean Wasserstein convergence; we are happy to mention them in the camera ready. Regarding your comment about assuming $\ell_f$ is in an RKHS, please see point 2 of our response to Reviewer #2.

**(b) Faster convergence:** The point you make about norms cancelling with rates is fair. We mention Wasserstein's $O(n^{-1/d})$ rate in the paper because it is relevant to the application of Principle 2.1. However, as discussed in point 1 of our response to Reviewer #2, we don't mean the remark about $O(n^{-1/2})$ vs $O(n^{-1/d})$ to claim the MMD results were always better. Instead, the different convergence properties of different distances motivates studying different DRO formulations. We will edit the paper to make this clearer.

[Meta-Review · NeurIPS 2019]

After thorough discussions among the area chair and reviewers, we concur that, albeit there remain several open questions, the paper provides a substantial contribution at the intersection of DRO and ML. Since the DRO has been neglected by the ML community despite its relevance in many ML applications, this work could potentially stimulate future work along this direction. Hence, I recommend that the paper gets accepted for publication at NeurIPS. Nevertheless, I would urge the authors, in the camera-ready version, to be candid about the limitations of their analysis and the need for future work. For example, the authors should explicitly mention the limitations of the loose upper bound in Theorem 3.1 as well as the fact that the constant M in Corollary 3.1 often depends on the dimension which is suboptimal.